# Analysis of Italian *BRCA1/2* Pathogenic Variants Identifies a Private Spectrum in the Population from the Bergamo Province in Northern Italy

**DOI:** 10.3390/cancers13030532

**Published:** 2021-01-30

**Authors:** Gisella Figlioli, Arcangela De Nicolo, Irene Catucci, Siranoush Manoukian, Bernard Peissel, Jacopo Azzollini, Benedetta Beltrami, Bernardo Bonanni, Mariarosaria Calvello, Davide Bondavalli, Barbara Pasini, Francesca Vignolo Lutati, Paola Ogliara, Monica Zuradelli, Valeria Pensotti, Giovanna De Vecchi, Sara Volorio, Paolo Verderio, Sara Pizzamiglio, Giuseppe Matullo, Serena Aneli, Giovanni Birolo, Federica Zanardi, Carlo Tondini, Alberto Zambelli, Luca Livraghi, Michela Franchi, Paolo Radice, Paolo Peterlongo

**Affiliations:** 1Genome Diagnostics Program, IFOM, FIRC Institute for Molecular Oncology, 20139 Milan, Italy; gisella.figlioli@ifom.eu (G.F.); irenecatucci@hotmail.it (I.C.); 2Cancer Genomics Program, Veneto Institute of Oncology IOV-IRCCS, 35128 Padua, Italy; arcangela.denicolo@iov.veneto.it; 3Unit of Medical Genetics, Department of Medical Oncology and Hematology, Fondazione IRCCS Istituto Nazionale dei Tumori, 20133 Milan, Italy; Siranoush.Manoukian@istitutotumori.mi.it (S.M.); bernard.peissel@istitutotumori.mi.it (B.P.); Jacopo.Azzollini@istitutotumori.mi.it (J.A.); benedetta.beltrami@unimi.it (B.B.); 4Division of Cancer Prevention and Genetics, IEO, European Institute of Oncology IRCCS, 20141 Milan, Italy; bernardo.bonanni@ieo.it (B.B.); mariarosaria.calvello@ieo.it (M.C.); davide.bondavalli@ieo.it (D.B.); 5Department of Medical Sciences, University of Turin, 10126 Turin, Italy; barbara.pasini@unito.it (B.P.); giuseppe.matullo@unito.it (G.M.); serena.aneli@gmail.com (S.A.); giovanni.birolo@gmail.com (G.B.); 6Medical Genetics Unit, University Hospital Città della Salute e della Scienza di Torino, 10126 Turin, Italy; fvignololutati@cittadellasalute.to.it (F.V.L.); pogliara@cittadellasalute.to.it (P.O.); 7Department of Oncology, IRCCS Humanitas Research Hospital, via Manzoni 56, Rozzano, 20089 Milan, Italy; monica.zuradelli@cancercenter.humanitas.it; 8Cancer Genetic Test Laboratory, Cogentech srl Società Benefit a Socio Unico, 20139 Milan, Italy; valeria.pensotti@cogentech.it (V.P.); giovanna.devecchi@cogentech.it (G.D.V.); sara.volorio@cogentech.it (S.V.); 9Unit of Bioinformatics and Biostatistics, Department of Applied Research and Technological Development, Fondazione IRCCS Istituto Nazionale dei Tumori, 20133 Milan, Italy; Paolo.Verderio@istitutotumori.mi.it (P.V.); Sara.Pizzamiglio@istitutotumori.mi.it (S.P.); 10Bioinformatics Core Unit, IFOM, FIRC Institute of Molecular Oncology, 20139 Milan, Italy; federica.zanardi@ifom.eu; 11Medical Oncology Unit, ASST Papa Giovanni XXIII, 24127 Bergamo, Italy; carlo.tondini@asst-pg23.it (C.T.); alberto.zambelli@asst-pg23.it (A.Z.); livraghi.luca@gmail.com (L.L.); mfranchi@fondazionefrom.it (M.F.); 12Department of Medical Biotechnologies, University of Siena, 53100 Siena, Italy; 13Unit of Molecular Bases of Genetic Risk and Genetic Testing, Department of Research, Fondazione IRCCS Istituto Nazionale dei Tumori, 20133 Milan, Italy; Paolo.Radice@istitutotumori.mi.it

**Keywords:** *BRCA1*, *BRCA2*, pathogenic variants, spectrum, breast cancer, Italy, Bergamo province

## Abstract

**Simple Summary:**

The Italian population is characterized by a high genetic heterogeneity mostly due to its long history of migration and colonization and to its geographical conformation. Consistently, several *BRCA1/2* pathogenic variants (PVs) have been reported to be recurrent or even founder in defined geographical areas including the Bergamo province in Northern Italy. In this study, we retrospectively analyzed the data from 1019 women affected with breast cancer with *BRCA1/2* PVs. We compared the *BRCA1/2* PVs spectrum found in the carrier individuals from the Bergamo province (BGP) with that of the general Italian population. We found that the majority of the BGP PVs had a local origin and remained confined to the BGP or to the surrounding Lombardy region. We also observed that the BGP *BRCA1/2* PV spectrum is private and conserved comprising a smaller number of variants with an average higher frequency with respect to that of carrier individuals from the rest of Italy.

**Abstract:**

Germline pathogenic variants (PVs) in the *BRCA1* or *BRCA2* genes cause high breast cancer risk. Recurrent or founder PVs have been described worldwide including some in the Bergamo province in Northern Italy. The aim of this study was to compare the *BRCA1/2* PV spectra of the Bergamo and of the general Italian populations. We retrospectively identified at five Italian centers 1019 *BRCA1/2* PVs carrier individuals affected with breast cancer and representative of the heterogeneous national population. Each individual was assigned to the Bergamo or non-Bergamo cohort based on self-reported birthplace. Our data indicate that the Bergamo *BRCA1/2* PV spectrum shows less heterogeneity with fewer different variants and an average higher frequency compared to that of the rest of Italy. Consistently, four PVs explained about 60% of all carriers. The majority of the Bergamo PVs originated locally with only two PVs clearly imported. The Bergamo *BRCA1/2* PV spectrum appears to be private. Hence, the Bergamo population would be ideal to study the disease risk associated with local PVs in breast cancer and other disease-causing genes. Finally, our data suggest that the Bergamo population is a genetic isolate and further analyses are warranted to prove this notion.

## 1. Introduction

Individuals carrying a germline pathogenic variant in the *BRCA1* or *BRCA2* gene have a significantly increased lifetime risk of developing breast and ovarian cancer [1]. Diagnostic screening performed worldwide based on gene sequencing has identified thousands of different *BRCA1/2* pathogenic variants (PVs). Although most of these PVs are individually rare, or even private to single families, recurrent variants, which sometimes could be traced to a common ancestor (founder effect), have been described within ethnically and/or geographically defined populations [2,3,4].

Traditionally, well-documented founder *BRCA1/2* PVs, such as those in the Ashkenazim [5,6], Icelanders [7,8], and Finns [9], have been instrumental to informed prioritization strategies for time- and cost-effective genetic testing and prompt identification of carrier individuals. With the advent of gene panel testing through next generation sequencing, genetic isolates, with their reduced genetic and environmental heterogeneity, are still a valuable resource to delve into genotype-phenotype associations of both Mendelian and complex disorders [10,11,12]. For instance, relating to variation in high-risk breast/ovarian cancer predisposing genes, refinement of penetrance estimates via elucidation of modifying gene-gene and gene–environment relationships would be a prerequisite to a more effective and personalized care of carrier individuals.

In Italy, population genetics analyses have brought to light a heterogeneous genetic background mostly imputable to repeated colonization and migratory waves and to an uneven geographical landscape, which has also facilitated the appearance of genetic isolates [13,14,15,16,17,18]. With respect to the *BRCA1/2* genes, a few PVs recurring in defined geographical areas and/or isolated populations have been described, e.g., in Calabria [19], Sardinia [20], Tuscany [21,22], Veneto [23], and Friuli-Venezia Giulia [24], reviewed in [3].

More recently, the Bergamo province in the Northern Italian region of Lombardy has emerged as a restricted geographical area inhabited by what appears to be a genetically homogenous population. We showed that the *BRCA1*:c.190T>C, is a founder variant in breast cancer families from Bergamo and has an estimated age of ~3000 years [25]. Upon extending our investigation to *PALB2*, the third high-penetrance breast cancer gene identified thus far, we reported that also the c.1027C>T PV has been exclusively identified in breast cancer patients—and to a significantly lower extent, in healthy individuals—from Bergamo [26,27].

To gain further insights into the *BRCA1/2* PV spectrum of the Bergamo province population, we retrospectively collected data from 1019 *BRCA1/2* PV carrier individuals ascertained at five Italian centers. We assigned them, based on self-reported birthplace, to the Bergamo province (BGP) or the non-Bergamo province (non-BGP) cohort, and carried out comparative analyses. We report herein a distinct distribution, frequency, and type of *BRCA1/2* PVs in the BGP compared to the general Italian population, consistent with the hypothesis that the BGP *BRCA1/2* PV spectrum is private.

## 2. Results

### 2.1. Distribution of the PVs Found in BGP

Twenty-five different BRCA1/2 PVs were identified in the 100 BGP carriers. Of these, 10 had a frequency ranging between 34% and 3%, and were deemed as common; the remaining 15 PVs were considered rare as they were found in one or two carriers (Table 1). All the 10 common PVs were found also outside the BGP. The BRCA1:c.190T>C, largely the most common BGP PV, was also found almost uniquely in several North-Italian regions; four other common PVs, namely the BRCA2:c.5904_5907delAGTC, BRCA1:c.2727_2730delTCAA, BRCA2:c.6469C>T, and BRCA2:c.6625dupA were found only in Lombardy. Conceivably, these five PVs may have originated in BGP and then spread throughout Lombardy and Northern Italy. The remaining five common PVs were found in BGP even though some carriers were observed in other regions (Table 1). Interestingly, four common PVs, namely BRCA1:c.190T>C, BRCA2:c.5904_5907delAGTC, BRCA2:c.8878C>T, and BRCA1:c.5062_5064delGTT were found in 59 of the 100 BGP carriers (Table 1; Appendix A). Thus, only four PVs explain more than half of the BGP BRCA1/2 carriers. Of the 15 rare PVs, 11 were almost exclusively found in BGP or in BGP and Lombardy. Carriers of the BRCA2:c.6468_6469delTC and BRCA2:c.1813delA were found in BGP but also in regions other than Lombardy. The BRCA1:c.5266dupC and BRCA1:c.514delC PVs were spread nation-wide with frequencies clearly suggesting that they originated from Apulia and Sicily, respectively. Overall, these observations indicate that the majority of BRCA1/2 PVs found in BGP originated locally—with a handful PVs explaining about 60% of the carriers—and suggest that the BRCA1/2 PV spectrum in BGP is private.

### 2.2. Comparison between the BGP and Non-BGP PV Spectra

In 1019 carriers, we identified a total of 304 different PVs; of these, 25 were found in BGP carriers and 299 in non-BGP carriers. Thus, 20 PVs were common to the BGP and non-BGP cohorts. We studied the PV distribution and frequency to test whether or not the BGP and non-BGP PV spectra were different. As a first test, we considered the number of different PVs with respect to the number of carriers in the two cohorts. Since 299 different PVs were observed in the 919 non-BGP carrier cohort, 32 PVs should have been proportionally found in the 100 Bergamo carriers. On the contrary, in this cohort, only 25 PVs were identified. Even if these numbers are not statistically different (X^2^
_(dof=1)_ = 1.28, *p* = 0.26), the Bergamo PV spectrum seems to comprise a smaller number of PVs. Consistently, four PVs account for approximately 59% of the BGP carriers while 50 PVs were necessary to explain 59% of the non-BGP carriers. We further studied the distribution of the 10 most frequent PVs in the non-BGP carriers (Table 2). None of these PVs had the same frequency in the BGP carriers. In particular, one PV was statistically less frequent in the BGP cohort with frequency difference of 8 (*p* = 0.0056). Five PVs were more frequent in the BGP carriers with three showing statistically significant higher frequency difference between 4.9 and 31.7, and two having frequency difference slightly higher than 2. The four remaining PVs, which altogether were found in 75/919 (8.2%) of the non-BGP carriers, were not found in the BGP carriers (Table 2). Finally, we studied the 20 PVs that were found in both the BGP and non-BGP carriers. We implemented a univariate logistic regression analysis to assess the strength of association between being carrier of a given PV and the BGP versus non-BGP birth sites. Figure 1 reports the odds ratios (ORs) and 95% confidence intervals (95% CI) in logarithmic scale. We observed that 10 PVs were statistically associated with the birth site. Of these PVs, nine were found to be positively associated with BGP birth site indicating that carriers of one of these nine PVs are more likely to be from BGP than those without PVs. Overall, by jointly considering the carriers of at least one of the 20 PVs their likelihood to be from BGP is about 37 times higher than those without PVs. Altogether, these data point out that the BGP BRCA1/2 PV spectrum is characterized by lower heterogeneity with respect to the non-BGP carriers. Moreover, the BGP and non-BGP PV spectra appear to be different in terms of PV distribution, type and frequency.

## 3. Discussion

In this study, we analyzed the *BRCA1/2* PVs found in individuals from the BGP and compared them with those found in individuals from the rest of Italy. Our data indicate that the majority of the BGP PVs had a local origin and remained confined to the BGP or to the surrounding Lombardy region, with only two PVs clearly “imported” from Southern Italy. We also showed that the number of different PVs found in BGP carriers was lower, albeit not significantly, than that in the carriers from the rest of the country. Four PVs accounted for the great majority (59%) of the BGP carriers. Moreover, we showed that some of the most common non-BGP PVs had a statistically different frequency in BGP. We also observed that the 20 PVs found in both non-BGP and BGP carriers were overall statistically associated with the BGP birth site. Specifically, among carriers of at least one of the 20 PVs, the likelihood to be from BGP is about 37 times higher than those carriers with other PVs. All these observations concur to highlight that BGP is characterized by a private *BRCA1/2* PV spectrum with lower genetic heterogeneity with respect to the rest of the country.

Data from the literature support our findings. Two recent studies reported the distribution of 471 *BRCA1/2* PVs found in the Middle East, North Africa, and South Europe [28] and of the *BRCA1/2* PVs found in more than 29,700 families collected worldwide [29]. Considering the distribution of the 25 BGP PVs in these two datasets, we observed that 19 PVs were found exclusively or prevalently in Italy, or were rarely found or not found at all. Of the remaining six PVs, two, namely *BRCA1*:c.5030_5033delCTAA and *BRCA1*:c.5266dupC were found in the majority of the countries in both datasets and four, namely *BRCA1*:c.3481_3491del11, *BRCA1*:c.1961dupA, *BRCA2*:c.7976+1G>A, and *BRCA2*:c.1813delA were found only in the dataset by Rebbeck et al. [29] (Table 1). These observations confirmed that 19 BGP PVs were originally from Italy and reinforced the speculation that 14 of these are only present in BGP and, possibly, in the surrounding Lombardy.

Other published studies documented the existence of PVs prevalent in the BGP. In particular, we previously studied the *BRCA1*:c.190T>C and found that it was common in the BGP and mapped to a conserved haplotype [25]. Interestingly this PV has now been shown to be associated with a risk for breast cancer, which is smaller than the originally estimated one (David Goldgar, manuscript in preparation). Similarly, the c.1027C>T (p.Gln343*) PV in the breast cancer predisposition gene *PALB2*, was identified in familial breast cancer cases from BGP with a carrier frequency of 5.3% [27] and located in a conserved haplotype [26]. Remarkably, the *PALB2*:1027C>T was never reported outside Italy ([30]; https://databases.lovd.nl/shared/genes/PALB2). Hence, BGP seems to be enriched with variants originating locally and is the ideal territory in which to further study the risk of breast cancer associated with specific PVs. Of course, other such variants could exist outside breast cancer genes. Consistently, a study on the frequency of the Y-chromosome haplogroups showed that individuals from the Bergamo valleys did not cluster with the other Italian populations [16].

Due to its geographical conformation and to its long history of migration and colonization, the Italian population is characterized by high genetic heterogeneity and by the presence of population isolates. Consistently, several *BRCA1/2* PVs have been shown to be recurrent, or even founder (i.e., in conserved haplotypes), in certain Italian regions. However, limited or no data exist on their recurrence in the rest of the country. Three PVs, namely *BRCA1*:c.116G>A, BRCA1:c.676delT, and *BRCA2*:c.7806-2A>G were reported to be confined within the Friuli Venezia Giulia region [24]. In our database, these three PVs were found in a total of seven carriers, four of whom were born in Friuli Venezia Giulia. A high prevalence rate of the *BRCA1*:c.4964_4982del19 was found in familial breast cancer cases from Calabria suggesting a possible founder effect [19]. We found this PV in 13 carriers, of whom six were born in Calabria and five in nearby regions. Two *BRCA2* founder variants, the *BRCA2*:c.8537_8538delAG and *BRCA2*:c.3723_3725delinsAT, were identified in Sardinia [20,31]; in our database, these two PVs were each found in one carrier from Sardinia. The *BRCA1*:c.3228_3229delAG and the *BRCA1*:c.3285delA PVs were described as originating from common ancestors in Tuscany [22]. A total of 11 individuals carried one of these PVs in our database, but only three were born in Tuscany. Finally, of the 16 individuals in the present study who carried the *BRCA1*:c.5062_5064delGTT [23,32] reported as a founder PV from Veneto [23], only two resulted from this region. Thus, our data indicate that the *BRCA1/2* PVs that were described to be recurrent in Friuli Venezia Giulia, Calabria, Tuscany, and Veneto are also diffused elsewhere on the national territory. Notably, we observed that the above-mentioned PVs, with the only exception of the *BRCA1*:c.4964_4982del19 from Calabria, were, in fact, not the most common but often among the rare PVs found in the carriers from those regions (Appendix A). Overall, these data do not corroborate the previous published studies describing these PVs as recurrent locally. We believe that these contradictory findings might be explained by bias due to different ascertainments. On one hand, monocentric studies might have been enriched by individuals who were ascertained as probands from different families but who were, in fact, related. On the other hand, we used the carrier’s birthplace as a proxy for the origin of the PVs, but we acknowledge that this is a surrogate of the geographical origin of a variant, which may, therefore, not be completely accurate [33]. For instance, a carrier might have inherited the PV from her parents or grandparents born elsewhere. This may be especially true for individuals recruited in Milan and Turin, cities that have a long history of internal migrations.

To our knowledge, no similar studies were published before and our observations are based on a relatively small number of individuals; hence, further analysis of larger similar datasets would be warranted to confirm our findings.

## 4. Materials and Methods

### 4.1. BRCA1/2 PV Carriers

In this study, we collected data from 1019 individual carriers of *BRCA1/2* PVs that were representative of the heterogeneous Italian population with an excess of carrier individuals from the BGP. As a proxy for the carrier’s geographical origin, we used the birthplace, specified by region, province, and municipality. The data included in this study were from individuals originally ascertained at five different centers. Of these, three—Istituto Nazionale dei Tumori (INT), Istituto Europeo di Oncologia (IEO), and Istituto Clinico Humanitas (ICH)—were located in Milan; one, Azienda Città della Salute e della Scienza di Torino (ACSS), in Turin, and one, Ospedale Papa Giovanni XXIII (OPG), in Bergamo. All 1019 individuals were Italian female breast cancer probands who were subjected to diagnostic *BRCA1/2* variant screening at Cogentech (serving INT, IEO, ICH, and OPG), and at ACSS. At Cogentech, 141 probands were tested using a combination of Denaturing High-Performance Liquid Chromatography (DHPLC) and Protein Truncation Test (PTT) and 672 probands by Sanger sequencing of all coding exons and adjacent intronic regions. At ACSS, 206 probands were tested by Sanger sequencing of all coding exons and adjacent intronic regions (in some cases conformational DHPLC analysis and Sanger sequencing of aberrant fragments or MLPA analysis in samples without small nucleotide variants were performed). At all five centers, the probands were considered eligible for *BRCA1/2* variant screening based on family history of breast or ovarian cancer, early onset of breast cancer, and tumor subtype; however, we acknowledge and could not control for slight differences in the eligibility criteria across centers. All probands included in the study: (i) received a breast cancer diagnosis, (ii) carried a *BRCA1/2* pathogenic or likely pathogenic variant according to the Evidence-based Network for the Interpretation of Mutant Alleles (ENIGMA) variant classification criteria (https://enigmaconsortium.org/wp-content/uploads/2020/08/ENIGMA_Rules_2017-06-29-v2_5_1.pdf), and (iii) were of Italian ethnicity. In total, data from 1019 Italian *BRCA1/2* PV carriers were included in this study. Of these, 100 were assigned to the BGP cohort and 919 to the non-BGP cohort. Two individuals carried both a *BRCA1* PV and a *BRCA2* PV; hence, 581 individuals carried a *BRCA1* PV and 440 carried a *BRCA2* PV (Table 3, Appendix A). All the data analyzed in this study were obtained from individuals who signed an informed consent to the use of their data for research purposes.

### 4.2. Heterogeneity of the BRCA1/2 PV Carriers

We wanted to analyze data from carriers of *BRCA1/2* PV that were representative of the Italian population heterogeneity. Hence, we collected data from carriers from three centers in Milan (INT, IEO, and ICH) and one in Turin (ACSS), which are two cities inhabited by a heterogenous population resulting from South-to-North immigration that started about 60 years ago. In any case, to verify whether or not these ascertained carriers were representative of the Italian population, we considered the number of *BRCA1/2* PV carriers by region of birth and derived the 1:100,000 rates with respect to the number of residents in each region registered on December 31, 2019 (https://www.tuttitalia.it/regioni/popolazione/; Table 4). For about two-thirds of the country regions (*n* = 13), the 1:100,000 rates ranged between 0.5 and 2.0 indicating that we collected data from a fairly proportional number of carriers. In few regions, we observed higher or lower rates. The higher rates observed in Lombardy and Piedmont were expected based on the study design. The high rates for Apulia and Aosta Valley may be due to migration flows, specifically from Apulia to Milan and Turin and from Aosta Valley to Turin. We observed fewer carriers from Lazio, Molise, and Umbria probably owing to the less significant migration from these regions to the Northern regions, where the ascertainment centers were located. As opposed to Milan and Turin, the city of Bergamo—the fifth ascertainment center—is not a major immigration destination and its province is populated by people, who were prevalently born locally. Thus, the high rate for the Bergamo province reflected the study design (Table 4).

### 4.3. Statistical Analyses

The Goodness of Fit Chi-Square Test was adopted to compare the distribution of observed *BRCA1/2* PVs in the BGP to the expected ones based on the number of PV detected in the general Italian population. Comparison between *BRCA1/2* PVs carrier frequencies in BGP and non-BGP cohorts was performed by estimating the frequency differences and their 95% CIs according to Newcombe method as recommended by Altman et al. [34] together with the *p*-value from chi-square or Fisher exact test, when appropriate. The association between the birthplace (BGP versus non-BGP) and the PV status was assessed by univariate logistic regression analyses.

All statistical analyses were carried out with the SAS software (Version 9.4.; SAS Institute Inc., Cary, NC, USA) by adopting an α value of 0.05.

## 5. Conclusions

In this study, we provide evidence indicating that the spectrum of *BRCA1/2* PV found in breast cancer cases from BGP is private compared to that of the rest of Italy. This conserved spectrum showed lower variant heterogeneity comprising a smaller number of different variants with an average higher frequency. Consistently, we report here for the first time that four frequent PVs explain about 60% of all *BRCA1/2* PV carriers from BGP. The BGP population is ideal to study further the risk effects of breast cancer predisposing PVs such as *BRCA1*:c.190T>C and *PALB2*:c.1027C>T. Finally, our data support the hypothesis that the BGP is inhabited by a genetically isolated population and further analyses are warranted to prove this possibility.

## Figures and Tables

**Figure 1 cancers-13-00532-f001:**
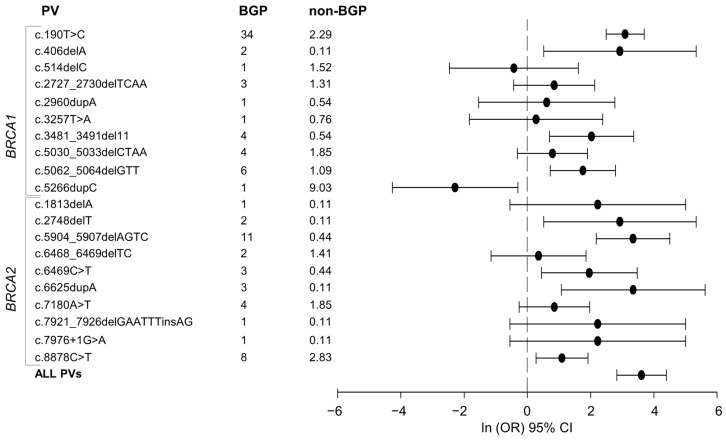
Ln odds ratio (ORs) and 95% confidence intervals (CIs) of the 20 PVs found in both the 100 BGP and the 919 non-BGP carriers. For each PV, the carrier frequency (%) is reported; the ln (OR) and the 95% CI are indicated by dots and horizontal lines, respectively. The ORs were estimated by univariate logistic model. The joint OR (ln OR = 3.61 (95% CI = 2.83–4.39) obtained by considering all the 20 PVs, is also showed.

**Table 1 cancers-13-00532-t001:** Distribution of the 25 Bergamo province (BGP) pathogenic variants (PVs).

PVs	Total No. of Carriers	Carrier Distribution (No. of Carriers; Freq%)		Suggested Origin
		BGP	Regions	
**Common (10)**				
*BRCA1*:c.190T>C	55	(34; 34.0)	Molise (1; 100), Trentino Alto Adige (1; 11.1), Veneto (2; 8.3), Friuli Venezia Giulia (1; 7.1), Lombardy non-BGP (14; 4.8), Piedmont (2; 1.2)	BGP ^1^
*BRCA2*:c.5904_5907delAGTC	15	(11; 11.0)	Lombardy non-BGP (4; 1.4)	BGP ^1^
*BRCA1*:c.2727_2730delTCAA	15	(3; 3.0)	Lombardy non-BGP (12; 4.1)	BGP/Lombardy ^1^
*BRCA2*:c.6469C>T	7	(3; 3.0)	Lombardy non-BGP (4; 1.4)	BGP ^1^
*BRCA2*:c.6625dupA	4	(3; 3.0)	Lombardy non-BGP (1; 0.3)	BGP ^1^
*BRCA2*:c.8878C>T	34	(8; 8.0)	Liguria (2; 12.5), Marche (1; 8.3), Lombardy non-BGP (15; 5.1), Veneto (1; 4.2), Piedmont (6; 3.6), Emilia Romagna (1; 2.4)	BGP/Liguria ^1^
*BRCA1*:c.5062_5064delGTT	16	(6; 6.0)	Trentino Alto Adige (2; 22.2), Veneto (2; 8.3), Lombardy non-BGP (5; 1.7), Piedmont (1; 0.6)	BGP/Trentino Alto Adige ^1^
*BRCA1*:c.3481_3491del11	9	(4; 4.0)	Emilia Romagna (3; 7.3), Lombardy non-BGP (2; 0.7)	BGP/Emilia Romagna ^2^
*BRCA1*:c.5030_5033delCTAA	21	(4; 4.0)	Liguria (3; 18.8), Piedmont (6; 3.6), Sicily (2; 3.2), Emilia Romagna (1; 2.4), Lombardy non-BGP (5; 1.7)	BGP/Liguria ^2^
*BRCA2*:c.7180A>T	20	(4; 4.0)	Friuli Venezia Giulia (1; 7.1), Lombardy non-BGP (12; 4.1), Piedmont (3; 1.8)	BGP/Friuli Venezia Giulia ^1^
**Rare (15)**				
*BRCA1*:c.406delA	3	(2; 2.0)	Lombardy non-BGP (1; 0.3)	BGP ^1^
*BRCA2*:c.2748delT	3	(2; 2.0)	Lombardy non-BGP (1; 0.3)	BGP ^1^
*BRCA2*:c.5473dupG	2	(2; 2.0)	None	BGP ^1^
*BRCA2*:c.8085_8086insGG	2	(2; 2.0)	None	BGP ^1^
*BRCA1*:c.53T>C	1	(1; 1.0)	None	BGP ^1^
*BRCA1*:c.1961dupA	1	(1; 1.0)	None	BGP ^2^
*BRCA1*:c.2960dupA	6	(1; 1.0)	Lombardy non-BGP (5; 1.7)	BGP/Lombardy ^1^
*BRCA1*:c.3257T>A	8	(1; 1.0)	Lombardy non-BGP (5; 1.7), Piedmont (2; 1.2)	BGP/Lombardy ^1^
*BRCA2*:c.7921_7926delGAATTTinsAG	2	(1; 1.0)	Lombardy non-BGP (1; 0.3)	BGP ^1^
*BRCA2*:c.7976+1G>A	2	(1; 1.0)	Lombardy non-BGP (1; 0.3)	BGP ^2^
*BRCA2*:c.8021dupA	1	(1; 1.0)	None	BGP ^1^
*BRCA2*:c.6468_6469delTC	15	(2; 2.0)	Veneto (2; 8.3), Marche (1; 8.3), Tuscany (1; 4.3), Campania (3; 7.5), Apulia (3; 2.1), Piedmont (3; 1.8)	BGP/Campania ^1^
*BRCA2*:c.1813delA	2	(1; 1.0)	Basilicata (1; 11.1)	BGP/Basilicata ^2^
*BRCA1*:c.514delC	15	(1; 1.0)	Sicily (8; 12.7), Lombardy non-BGP (4; 1.4), Apulia (1; 1.0), Piedmont (1; 0.6)	Sicily ^1^
*BRCA1*:c.5266dupC	84	(1; 1.0)	Apulia (31; 30.1), Abruzzo (2; 15.4), Trentino Alto Adige (1; 11.1), Sicily (5; 7.9), Calabria (3; 7.7), Campania (3; 7.5), Lombardy non-BGP (22; 7.5), Piedmont (12; 7.2), Friuli Venezia Giulia (1; 7.1), Liguria (1; 6.3), Emilia Romagna (2; 4.9)	Apulia ^2^

^1^ Nineteen PVs found exclusively or prevalently in Italy, or rarely found or not found at all in previously published *BRCA1/2* PV distribution studies (Laitman et al.; Rebbeck et al. see reference list). ^2^ Six PVs found in many countries as reported in one or both of the two distribution studies.

**Table 2 cancers-13-00532-t002:** The 10 most common non-BGP PVs and their frequency difference in the BGP and non-BGP carriers.

Non-BGP PV	Non-BGP Carriers (%)	BGP Carriers (%)	BGP—Non-BGP Frequency Difference (95%CI)	*p*-Value
*BRCA1*:c.5266dupC	83 (9.0)	1 (1.0)	−8 (−10.2 to −3.3)	0.0056
*BRCA1*:c.190T>C	21 (2.3)	34 (34.0)	31.7 (23.1 to 41.5)	<0.0001
*BRCA1*:c.5062_5064delGTT	10 (1.1)	6 (6.0)	4.9 (1.6 to 11.4)	0.0002
*BRCA2*:c.8878C>T	26 (2.8)	8 (8.0)	5.2 (1.1 to 12.2)	0.0063
*BRCA2*:c.7180A>T	16 (1.7)	4 (4.0)	2.3 (−0.4 to 8.1)	0.1219
*BRCA1*:c.5030_5033delCTAA	17 (1.8)	4 (4.0)	2.2 (−0.5 to 8)	0.1509
*BRCA1*:c.1088delA	25 (2.7)	0 (0.0)	−2.7 (−4 to 1.1)	0.0951
*BRCA1*:c.181T>G	20 (2.2)	0 (0.0)	−2.2 (−3.3 to 1.6)	0.1361
*BRCA2*:c.5796_5797delTA	15 (1.6)	0 (0.0)	−1.6 (−2.7 to 2.1)	0.1986
*BRCA1*:c.3257T>G	15 (1.6)	0 (0.0)	−1.6 (−2.7 to 2.1)	0.1986

**Table 3 cancers-13-00532-t003:** Number of carriers of *BRCA1* or *BRCA2* PVs included in the study by testing ascertainment center and BGP and non-BGP cohort.

Testing-Ascertainment Center	All Carriers	Carriers of *BRCA1* PVs (%)	Carriers of *BRCA2* PVs (%)
Cogentech-IEO	353	195 (55.2)	158 (44.8)
Cogentech-INT	309	187 (60.5)	122 (39.5)
Cogentech-OPG *	90	53 (58.9)	38 (41.1)
Cogentech-ICH	61	39 (63.9)	22 (36.1)
ACSS *	206	107 (51.9)	100 (48.1)
All cohorts	1019 *	581 (57.0)	440 (43.0)
BGP	100	59 (59.0)	41 (41.0)
non-BGP *	919	522 (56.8)	399 (43.2)
All	1019 *	581 (57.0)	440 (43.0)

* Of the 1019 carriers, one from OPG and one from ACSS (both from non-BGP) carried each a PV in *BRCA1* and a PV in *BRCA2*.

**Table 4 cancers-13-00532-t004:** Analysis of the *BRCA1/2* PV carriers’ heterogeneity.

Region	Region of Birth of Carrier Ascertained in				Number of Residents ^#^	Rate * (1:100,000)
	All Cities	Milan (INT, IEO, ICH)	Turin (ACSS)	Bergamo (OPG)		
Lombardy (Non-BGP)	293	281	3	9	8,987,585	3.3
BGP	100	29	0	71	1,116,384	9.0
Lazio	20	18	2	0	5,865,544	0.3
Campania	40	32	7	1	5,785,861	0.7
Sicily	63	50	9	4	4,968,410	1.3
Veneto	24	19	4	1	4,907,704	0.5
Emilia Romagna	41	35	5	1	4,467,118	0.9
Piedmont	167	43	124	0	4,341,375	3.9
Apulia	103	85	17	1	4,008,296	2.6
Tuscany	23	22	1	0	3,722,729	0.6
Calabria	39	27	12	0	1,924,701	2.0
Sardinia	22	17	3	2	1,630,474	1.4
Liguria	16	13	3	0	1,543,127	1.0
Marche	12	11	1	0	1,518,400	0.8
Abruzzo	13	10	3	0	1,305,770	1.0
Friuli Venezia Giulia	14	8	6	0	1,211,357	1.2
Trentino Alto Adige	9	9	0	0	1,074,819	0.8
Umbria	3	3	0	0	880,285	0.3
Basilicata	9	6	3	0	556,934	1.6
Molise	1	1	0	0	302,265	0.3
Aosta Valley	7	4	3	0	125,501	5.6
All regions (non-BGP)	919	694	206	19	59,128,255	1.6
All regions	1019	723	206	90	60,244,639	1.7

^#^ Derived from https://www.tuttitalia.it/regioni/popolazione/ on 31 December 2019. * Rates were derived considering the number of *BRCA1/2* PV carriers by region of birth and the number of residents in each region.

## Data Availability

The data presented in this study are available in Appendix A.

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
