# Peer review of "Analysis of Italian BRCA1/2 Pathogenic Variants Identifies a Private Spectrum in the Population from the Bergamo Province in Northern Italy"

_cancers, 2021, doi:10.3390/cancers13030532_

Round 1
Reviewer 1 Report
The proposed manuscript by Figlioli et al. presents the results from a retrospective study aimed to compare the spectrum of BRCA1/2 pathogenic variant in the genetically homogenous population of Bergamo province in Italy to those in the general Italian population. Study involved total of 1,019 women with breast cancer with BRCA1/2 genetic variants stratified into two groups – with Bergamo or non-Bergamo origin. Results from the analysis demonstrated lower heterogeneity, fewer different variants, and higher frequency of the BRCA1/2 PV spectrum in the Bergamo cohort, compared to the non-Bergamo one. Based on the obtained findings, the authors of the study concluded that the present data provide additional support the hypothesis that the Bergamo population is genetically isolated.
The study is designed and conducted correctly and manuscript is written well. I did not find any problems with the statistics or data interpretations. My following comment is of minor importance and aims to improve the quality of the manuscript.
Specific comments and recommendations:
- Discussion: My recommendation to the authors is in one paragraph to highlight the strengths and the limitations of their study compared to previous similar studies.
Author Response
Dear Reviewer,
Thank you for your revision. We have addressed all of your comments. Specifically, We have added, at the end of the Discussion section, the following text: "To our knowledge, no similar studies were published before and our observations are based on a relatively small number of individuals; hence, further analysis of larger similar data-sets would be warranted to confirm our findings."
Reviewer 2 Report
In the manuscript entitled “Analysis of Italian BRCA1/2 pathogenic variants identifies a private spectrum in the population from the Bergamo province in Northern Italy" the authors analyzed the BRCA1/2 pathogenic variants (PVs) found in individuals from the Bergamo province (BGP) and compared them with those found in individuals from the rest of Italy. The authors reported that four frequent PVs explain about 60% of all BRCA1/2 PV 276 carriers from BGP. The BGP population is ideal to study further the risk effects of breast cancer predisposing 277 PVs such as BRCA1:c.190T>C and PALB2:c.1027C>T.
The study is interesting and rich in contents. The supplementary table is clear.
We considered suitable for publication in CANCERS.
Author Response
Dear Reviewer,
Thank you for your revision.
Reviewer 3 Report
This is an epidemiological study. Although the interesting data are well –presented, there is no cellular, molecular, and/or functional studies. To my opinion, it should be published in a more specialized epidemiological journal.
Author Response
Dear Reviewer,
Thank your for you revision. We agree that this is an epidemiological study with no cellular, molecular, and/or functional data. However, to our knowledge, “Cancers” aims at publishing also articles based on sole epidemiological data.
Reviewer 4 Report
Please explain briefly what the Newcombe method is and why it was chosen.
Line 269: "The association between the birth place (BGP vs non-BGP) and the PVs status was assessed by resorting to a univariate logistic regression analysis." I would remove 'resorting to'.
Does figure 1 show odds ratios or logORs? And are they (log)ORs or being from Bergamo province?
Author Response
Dear Reviewer,
Thank your for you revision. We have addressed all of your comments. Specifically:
Q: Please explain briefly what the Newcombe method is and why it was chosen.
R: The confidence interval of frequencies difference was calculated according to the Newcombe method that is the recommended one given by Altman et al. (2000, [see ref. 34]) according to a comparative study published on Statistics in Medicine (Interval estimation for the difference between independent proportions: comparison of eleven methods. Stat Med 1998;17:873-90). Briefly, starting from the lower and upper confidence limit of the compared frequencies (p1 and p2) denoted as L1,U1 and L2,U2, the Newcombe confidence limits (LD and UD) for the frequencies difference (D=p1-p2) are computed as: LD = D- [squareroot((p1-L1)2+(U2-p2)2 ) and UD = D+ [squareroot((p2-L2)2+(U1-p1)2)
Q: Line 269: "The association between the birth place (BGP vs non-BGP) and the PVs status was assessed by resorting to a univariate logistic regression analysis." I would remove 'resorting to'.
R: We have removed “resorting to” from the text
Q: Does figure 1 show odds ratios or logORs? And are they (log)ORs or being from Bergamo province?
R: we have clarified in the Figure 1 legend that are reported the lnORs from Bergamo province.